# OpenReview forum: "Bias-Augmented Consistency Training Reduces Biased Reasoning in Chain-of-Thought"
_ICLR.cc/2025/Conference — ICLR 2025 Conference Withdrawn Submission_

### Official Review · Reviewer_mzdz · 2024-11-01

**Soundness:** 3
**Presentation:** 4
**Contribution:** 3
**Rating:** 6
**Confidence:** 3

**Summary:**

This paper introduces bias-augmented consistency training (BCT), an unsupervised fine-tuning method to reduce biased reasoning in chain-of-thought (CoT) language model outputs. The authors demonstrate that BCT significantly reduces biased reasoning by training models to give consistent reasoning across prompts with and without biasing features. The key contribution is showing that training on a single bias type generalizes well to reducing other forms of biased reasoning and to new tasks, without requiring labeled data. The authors also validate that BCT minimally impacts model performance on standard tasks.

**Strengths:**

1. The introduced bias-augmented consistency training is simple yet effective. Its unsupervised nature suggests broad applicability and scalability.
2. The authors presents strong empirical results showing generalization across bias types and tasks.
3. The paper is very well-written and easy to follow.

**Weaknesses:**

1. My main concern with the proposed BCT is its potential negative impact on few-shot performance or, more broadly, in-context learning. While Appendix I shows some evidence for this, I'm not fully convinced, as Figure 9 only includes TruthfulQA results. I recommend including few-shot and few-shot CoT results on additional benchmarks, such as MMLU, to provide a more comprehensive assessment.
2. Section 5.2 demonstrates that BCT reduces coherent biased reasoning, highlighting a trade-off between CoT faithfulness and sensitivity to biases. This important finding warrants further discussion and analysis.
3. Some experimental design choices lack clear justification. For instance, the rationale behind measuring task and bias generalization simultaneously, as well as the selection of training data mixture proportions, could be better explained.

**Questions:**

1. How does the instruction-following data influence the effectiveness of BCT?
2. Will BCT affect the model's calibration?

---

### Official Review · Reviewer_JNUp · 2024-11-03

**Soundness:** 3
**Presentation:** 3
**Contribution:** 2
**Rating:** 5
**Confidence:** 4

**Summary:**

This paper proposes a new method called bias-augmented consistency training to improve faithfulness in CoT Reasoning. Specifically, the paper shows that training on certain bias types can help reduce biases in other types of biased reasoning behaviors.

**Strengths:**

- The studied problem is important: faithfulness in CoT reasoning is relatively under-explored and it's important to ensure the models to produce unbiased reasoning to users.

- The results on generalization to other types of biased reasoning is very interesting, and demonstrates the usefulness of the proposed method.

- The authors provided fairly comprehensive analysis on models' behavior when trained with BCT, especially the observation where BCT reduces coherent biased reasoning is kind of interesting.

**Weaknesses:**

- Novelty: overall the proposed method is similar to existing literature on reducing bias in ML/NLP models with data augmentation or counterfactual augmentation, hence the novelty of the work is a bit lacking. The authors should provide more discussion on how this work is different from existing literature.

- For the generalization behavior, currently the main experiments are done on Sycophancy examples and generalization is shown on other bias types. It would be more useful to show if this trend holds or not (and to what extent) regardless of which bias type the model is being trained on. Some of the experiments are provided in Appendix E but the results are not very detailed. In Figure 5, why would multi-bias training perform on-par and sometimes even worse than single-bias? More discussion on this would be useful.

Minor:
- Many of the important results are included in the appendix, it would be more useful to re-organize the experiment section a bit and show all the major results in the main text. For example, the main results are only shown over one model: gpt-3.5-turbo-0613. The authors provided some additional experiments on LLaMa3-8B in the appendix, but it would be more useful to bring those experiments to the main text to show that the method generalizes across models. Also, is there a study on what model sizes are required for BCT to be effective?

**Questions:**

- What is the BCT training objective? From the description in Section 2.2, is it simply adding both biased and unbiased responses when fine-tuning the model, or the objective actually changed?

---

### Official Review · Reviewer_VK7Y · 2024-11-03

**Soundness:** 2
**Presentation:** 3
**Contribution:** 2
**Rating:** 3
**Confidence:** 3

**Summary:**

This paper proposes a method for fine-tuning of large language models by using the concept of biased-consistency training (BCT). The aim is to reduce biased reasoning in chain-of-thought by incorporating a suite testing nine forms of biased reasoning under the (BCT) unsupervised fine-tuning method. The method is tested on question answering tasks run with GPT-3.5T.

**Strengths:**

- Originality: The paper addresses an important problem of doing biased/unbiased reasoning answers to questions that are potentially biased.
- Quality/Clarity: The paper is clearly written and the structure is simple and clear to follow.
- Significance: The paper provides some results that show that, under standard bias metrics, the proposed approach could help generate unbiased answers in the context of biased questions.

**Weaknesses:**

- From Figure 2, I understand the main contribution of the paper is the BCT fine-tuning objective. However, there is no comprehensive explanation of what it entails, the assumptions, limitations of the proposed approach. Furthermore, Section 2 which is supposed to be the methodology part delves more around existing approaches, whilst leaving a single small paragraph for the explanation of the method.
- The experiments are limited to GPT-3.5T. Therefore, there is no demostration of how generalizable the proposed method is to LLMs in general.
- Beyond Figure 3, there is no other place in the experiments where the results of the proposed method in question answering tasks are clearly explained, with clear examples that show how this proposed method produces unbiased/biased answers.

**Questions:**

- Are there results for other LLMs beyond GPT-3.5T?
- Can you formalize what the BCT objective is?

---

### Official Review · Reviewer_k94o · 2024-11-03

**Soundness:** 3
**Presentation:** 3
**Contribution:** 2
**Rating:** 3
**Confidence:** 3

**Summary:**

This paper introduces Bias-Augmented Consistency Training (BCT), presenting a novel unsupervised approach to reducing biased reasoning in large language models. The work reframes the challenge of biased reasoning as a consistency problem, training models to maintain coherent reasoning patterns across prompts with and without biasing features. Through extensive evaluation the authors demonstrate BCT's effectiveness in trained bias and held-out biases.

**Strengths:**

The paper addresses an important problem in AI systems - the tendency of language models to generate biased reasoning without acknowledging the influence of biasing features. The empirical results show some promise, with reported reductions in certain types of biases. The experimental setup is generally well-documented, facilitating reproducibility, and the authors are commendably transparent about some of the method's limitations.

**Weaknesses:**

While the presented approach to bias mitigation is interesting, several weaknesses limit its potential contribution.

- The paper lacks a clear theoretical explanation for the effectiveness of BCT, making it difficult to predict its applicability to different biases. While this is understandably a common challenge for LLMs, the resource-intensive nature of BCT, requiring numerous paraphrases for each bias type, raises concerns about its scalability and practicality. Additionally, the paper does not adequately address the potential challenges in generating high-quality paraphrases at scale.

- The inconsistent performance of BCT across different bias types raises questions about its reliability and generalizability. Furthermore, the observed degradation in instruction-following capabilities suggests potential unintended consequences, highlighting the need for a more thorough investigation of potential negative impacts. Perhaps its worthwhile to be more comprehensive in more instruction-following tasks.

- The experimental evaluation is limited in scope, focusing primarily on multiple-choice questions and bias reduction metrics. This narrow focus raises concerns about the generalizability of the findings to open-ended tasks and real-world applications. The paper's claim of successful generalization to unseen biases lacks a convincing analysis of the underlying mechanisms.

- The paper does not adequately position its contribution within the broader context of existing debiasing approaches. While BCT demonstrates some empirical success, its limitations and potential drawbacks raise questions about its overall significance and potential advantages over alternative methods.

**Questions:**

Could you explain the mechanism behind cross-bias generalization, e.g., why does training on sycophancy bias help reduce pattern matching biases, or have you identified common features among biases that show better generalization?

The paper demonstrates BCT works but doesn't fully explain why. Could you provide analysis of what the model is actually learning during BCT?

Regarding paraphrase requirements, what is the minimum number of paraphrases needed for effective BCT? What is the relationship between number of paraphrases and performance?

Is it possible to experiment with some methods of balancing bias reduction with instruction following?

---

### Note · Authors · 2024-11-21

**Comment:**

We thank all the reviewers' comments and appreciate their valuable time. We work on continually improving this work.

**Withdrawal Confirmation:**

I have read and agree with the venue's withdrawal policy on behalf of myself and my co-authors.